Isolation and characterization of microsatellite markers for Sturnira parvidens and cross-species amplification in Sturnira species

Gutiérrez Edgar G. 1
Hernández Canchola Giovani 2
León Paniagua Livia S. 2
Martínez Méndez Norberto 1
Ortega Jorge artibeus2@aol.com 1
1 Department of Zoología, Instituto Politécnico Nacional/ENCB , CDMX , CDMX , México
2 Department of Biología Evolutiva, Facultad de Ciencias, UNAM , CDMX , CDMX , México
Goss Erica
Electronic publication date: 2017 May 24
Publication date: 2017
Volume: 5
Electronic Location ID: e3367
Received 2017 Mar 15; Accepted 2017 Apr 28
Copyright: ©2017 Gutiérrez et al.
Copyright year: 2017
Copyright holder: Gutiérrez et al.
License: This is an open access article distributed under the terms of the Creative Commons Attribution License, which permits unrestricted use, distribution, reproduction and adaptation in any medium and for any purpose provided that it is properly attributed. For attribution, the original author(s), title, publication source (PeerJ) and either DOI or URL of the article must be cited.
License URL: https://creativecommons.org/licenses/by/4.0/

Keywords: Microsatellites, Sturnira parvidens, Pal_finder, Illumina

Funding: Proyecto Ciencia Básica CONACyT 239482 Proyecto de Investigación SIP IPN 20160744 This work was granted by Proyecto Ciencia Básica CONACyT (239482) and Proyecto de Investigación SIP IPN (20160744). The funders had no role in study design, data collection and analysis, decision to publish, or preparation of the manuscript.

==============================
Background

Sturnira is one of the most species-rich genera in the Neotropics, and it is found from Mexico and the Lesser Antilles to Argentina. This genus forms a well-supported monophyletic clade with at least twenty-one recognized species, as well as several others under taxonomic review. Sturnira parvidens is a widespread frugivorous bat of the deciduous forests of the Neotropics, is highly abundant, and is a major component in fruit dispersal to regenerate ecosystems.

Methods

We used a technique based on Illumina paired-end sequencing of a library highly enriched for microsatellite repeats to develop loci for S. parvidens. We analyzed millions of resulting reads with specialized software to extract those reads that contained di-, tri-, tetra-, penta-, and hexanucleotide microsatellites.

Results

We selected and tested 14 polymorphic (di, tri, and tetra) microsatellites. All markers were genotyped on 26 different individuals from distinct locations of the distributional area of S. parvidens. We observed medium—high genetic variation across most loci, but only 12 were functionally polymorphic. Levels of expected heterozygosity across all markers were high to medium (mean HE = 0.79, mean HO = 0.72). We examined ascertainment bias in twelve bats of the genus, obtaining null/monomorphic/polymorphic amplifications.

Discussion

The Illumina paired-end sequencing system is capable of identifying massive numbers of microsatellite loci, while expending little time, reducing costs, and providing a large amount of data. The described polymorphic loci for S. parvidens in particular, and for the genus in general, could be suitable for further genetic analysis, including taxonomic inconsistencies, parentage/relatedness analysis, and population genetics assessments.

Introduction

The yellow-shouldered Mesoamerican bat (Sturnira parvidens) is primarily associated with lower elevations (0 to 2,000 m), and is found mainly in tropical/subtropical habitats and ecotones (Villalobos & Valerio, 2002). S. parvidens is found from the northern Mexican Pacific Slope and the northern Mexican Gulf Slope southward to Northern Costa Rica, and including the Yucatan Peninsula (G Hernández-Canchola & L León-Paniagua, 2017, unpublished data). S. parvidens has been caught in the understory and subcanopy of tropical and subtropical forests, in xeric scrubs, and in secondary and temperate forests. They are commonly found roosting in the foliage of forests of advanced successional stages, but their home ranges include mature and secondary forest (Evelyn & Stiles, 2003). They mainly consume fruit from plants representing early stages of plant succession, like pioneer trees (Cecropia peltata), pioneer herbs (Solanum americanum, S. torvun, S. ochraceo-ferrugineum, Capsicum annuum), or pioneer shrubs (Piper hispidum, P. lapathifolium; Olea-Wagner et al., 2007). This frugivorous species is an important seed disperser, carrying out an important ecosystemic role in the restoration of secondary tropical forests. It is considered abundant but, as fragmentation intensifies, the species is particularly vulnerable to local extinction (Evelyn & Stiles, 2003).

Pleistocene climatic oscillations and the complex orogeny of its distributional area shaped the phylogeography of this bat, generating two lowland lineages. The two genetic lineages, one in the Western Slope region of Mexico, and the other in the Eastern Slope region of Mexico and Central America, diverged into haplogroups around c. 0.423 Ma, and demographic expansion was detected later, after the splitting event (G Hernández-Canchola & L León-Paniagua, 2017, unpublished data). Sturnira is the most speciose genus of frugivorous bats. Due to its ability to colonize new areas, it adapted to produce complex groups showing different genetic lineages (Velazco & Patterson, 2013; Velazco & Patterson, 2014; G Hernández-Canchola & L León-Paniagua, 2017, unpublished data). The genus Sturnira involves a highly diversified and complex group of species. This speciose group of bats inhabits the entire Neotropic realm and includes three mountain basal species: S. aratathomasi, S. bidens, and S. nana. Also, it has been described as a clade formed by species that usually inhabit highland mountain forests: S. bogotensis, S. burtonlimi, S. erythromos, S. hondurensis, S. koopmanhilli, S. ludovici, S. magna, S. mordax, S. oporaphilum, S. perla, S. tildae and S. adrianae (Velazco & Patterson, 2013; Molinari et al., 2017). Lastly, it includes a group of species that inhabit lowland tropical forests: S. angeli, S. bakeri, S. lilium, S. luisi, S. new species 3, S. paulsoni, and S. parvidens (Velazco & Patterson, 2013).

No microsatellite molecular markers are known for Sturnira parvidens; our goal was to isolate and characterize polymorphic microsatellite loci for the species by using Next-Generation Sequencing. The development of these markers can be useful for understanding the genetic structure of subpopulations in its distributional range. They can be used to identify the impact of humans on the fragmentation of the populations and assess the divergent lineages formed by genetic drift. They can also be used to evaluate movements of individuals in the mosaic-fragmented landscapes, and discern the genetic component in the social structure of the population by assessing relatedness and paternity. We showed cross-species amplification in twelve species of the Sturnira genus, under the hypothesis of having a positive ascertainment bias due to the phylogenetic relatedness among species (Crawford et al., 1998; Li & Kimmel, 2013). Suitable cross-species amplification will facilitate studies in Sturnira related bat populations of Middle and South America.

Materials and Methods

We obtained tissue samples from 26 distinct individuals of S. parvidens from different localities in its distributional range in Mexico. Specimens were provided by Colección de Mamíferos del Museo de Zoología “Alfonso L. Herrera”, Facultad de Ciencias-Universidad Nacional Autónoma de México. Tissue samples were stored individually in 95% ethanol until analysis. We followed the guidelines set forth by the American Society of Mammalogists for the use of wildlife (Gannon, Sikes & Animal Care and Use Committee of the American Society of Mammalogists, 2007). Fieldwork was conducted with the permission of SEMARNAT (Secretaría del Medio Ambiente y Recursos Naturales de Mexico—permit FAUT-0307). Six samples were sent to the Savannah River Ecology Laboratory, for an enrichment library process. The facility follows their own protocol and provides a database of the resulting microsatellites. Meanwhile the rest of the specimens were used to standardize protocols and assess polymorphism in microsatellites.

DNA was extracted following the instructions of the Qiagen protocol (Blood and Tissue Kit, Cat No. 69504; Qiagen, Hilden, Germany) for shotgun sequences, and we used the Universal Salt Protocol to extract DNA from the remaining specimens (Aljanabi & Martinez, 1997). An Illumina paired-end shotgun library was prepared by shearing 1l g of tissue DNA using a Covaris S220 and following the standard protocol of the Illumina TruSeq DNA Library Kit. Five million of the resulting reads were analyzed with the program PAL_ FINDER_v0.02.03 (Castoe et al., 2012), in order to extract those reads that contained di-, tri-, tetra-, penta-, and hexanucleotide microsatellites.

Once positive reads were identified in PAL_FINDER, they were batched to a local installation of the program MSATCOMMANDER v 0.8.2 for primer design (Faircloth, 2008). We recovered 6,790 unique loci (48 hexa, 97 penta, 1,260 tetra, 1,097 tri and 4,288 dinucleotide—Fig. 1), but only 14 were chosen for PCR trials that were performed in a MultiGene™ Gradient Thermal Cycler (Labnet, Edison, NJ, USA). We directly labelled forward primers (FAM) for each of the chosen loci. PCR reactions were performed in a 10 µl volume containing 30 ng of DNA, 0.2 mM of dNTPs, 10 mM of each primer, 1 Taq buffer (Buffer PCR 10×), 0.3 µL MgCl2 (25 mM), and 1.0 U of FlexiTaq polymerase. PCR cycling conditions were as follows: initial denaturation at 95 °C for 3 min; followed by 30 cycles of 95 °C for 3 min, gradient temperature (ranging from 56 to 60 °C) for 30 s, and 72 °C for 2 min; extension of 68 °C for 8 min; and final ending of 4 °C. Exact annealing temperatures for each primer are given in Table 1. We visualized the PCR products by electrophoresis on 1.5% agarose gels. Markers were tested for amplification success, polymorphism and specificity in 26 individuals of S. parvidens.

Figure 1 Potentially amplifiable loci (PAL’s) with positive microsatellites found in the enriched library.

Perfect, imperfect and compound loci separated out for dinucleotide to hexanucleotide microsatellite forms.

Table 1 Primer sequences and characteristics of the 14 microsatellite loci isolated for Sturnira parvidens.

Annealing temperature not obtained (X).

Locus	Primer (Forward) (5–3′)	Primer (Reverse) (5–3′)	Motif	Annealing T (°C)	
Spar01	6 FAM-TGCCCTGAAGAACTTTGAGC	CCCATACTTCTCCCTCACAGC	AAAG(92)	58	
Spar02	6 FAM-AGAAAGAAAGGGAGGGCGG	TTCTTTATGCCCTTTGCTCTAGG	AAAG(104)	60	
Spar05	6 FAM-TGCCTGCCTAGTCTGTCACC	AAGCAGTTCCCATCACATGC	ATC(33)	56	
Spar06	6-FAM-CCTGGGATGAAGTTTCTGACG	GAATAATGGGAATACCAGAATAAGACG	TTC(30)	×	
Spar07	6 FAM-CTCCCACGGACAATCAACG	CCCAGATTGCTGCCTCTCC	TGC(30)	56	
Spar08	6 FAM-GGAGTCTCCTTCATTAAGTGCC	GGATGTGTTGTGAAGATTGTGC	ATT(30)	56	
Spar09	6 FAM-AAGTCCATTTCAAGGCTGGG	CCCATCATACCCTCCTTTGC	AC(44)	60	
Spar010	6 FAM-TCTGGCCTGAGGTATTTGGG	ACTGTAGCCACTTCCCTGCC	AC(44)	60	
Spar011	6 FAM-AAGCCACTGCCTTGTGCC	GACTCTCTGGACATTGGCCC	TC(44)	60	
Spar012	6 FAM-GGGAGTGAATGAGAAAGATAAAGTCC	CTGTCATTGCATGGGTTGG	AC(44)	60	
Spar013	6 FAM-AAAGATTCCTGGAGATCATACCC	TGAATGTATCCTAGGGCGAGC	AC(42)	60	
Spar014	6-FAM-TTTCTCTCACTGTCTAACTCTGCC	AGTCCTGGCAGGTGTGTCC	TC(32)	×	
Spar030	6 FAM-AATGGCACCATATTATTCTACATAGG	CCGTTCTAGGCTCAGTTTCC	ATT(36)	60	
Spar040	6 FAM-GACTGAGACAATTGCTTGAGATAGC	GAGTTTCAGGGAGTATTTCAGTGC	ATC(33)	60	

The results of the microsatellite profiles were examined using GeneMarker® v. 2.4.2 (SoftGenetics, State College, PA, USA) and peaks were scored by hand. We obtained the number of homozygotes and heterozygotes by scoring data. We estimated the proportion of polymorphic loci and the average number of alleles per locus by using the GDA software (Lewis & Zaykin, 2001). We assessed the observed (HO) and the expected heterozygosity (HE), linkage disequilibrium, and Hardy–Weinberg proportions by using Genepop 4.2 (Rousset, 2008), and corroborated with Arlequin 3.5 (Excoffier, Laval & Schneider, 2003). We used MICROCHECKER to screen null alleles in each locus (Van Oosterhout et al., 2004). We measured polymorphic information content (PIC) with Cervus 3.0.7 (Kalinowski, Taper & Marshall, 2007).

We probed cross-species amplification in tissues of twelve species of the genus: S. hondurensis, S. burtonlimi, S. oporaphilum, S. mordax, S. tildae, S. erythromos, S. bogotensis, S. magna, S. new species 3, S. luisi, S. lilium, and S. bakeri (Supplemental Information 1). All polymorphic loci were tested in the mentioned species by using similar PCR conditions. We followed the ascertainment bias hypothesis of broad amplification in similar phylogenetic species (Schlötterer, 2000).

Results

We obtained a total of 6,790 potentially amplifiable loci (PALs), containing perfect, imperfect, and compound microsatellites (Fig. 1). Dinucleotide microsatellites were the most abundant (4,288), followed by tetra (1,260); hexa microsatellites were the least abundant in our readings (48). PCR reactions showed that of the 14 loci tested, two were non-specific or monomorphic, and only 12 loci were polymorphic such that we were able to get proper amplification (Table 1). Annealing temperature ranged from 56 to 60 °C.

We found moderate levels of allelic richness, with an average of 8.8 alleles per locus in the representative selection from the wide area of the distribution of Sturnira parvidens. Polymorphic information content (PIC) presented values above 0.5 showing a significant content of alleles per locus. Allele frequencies showed a remarkable diversity of alleles per locus, driving a superior number of valuable loci to be used in different genetic analyses (Supplemental Information 2). No evidence of linkage disequilibrium was found on the analyzed loci. We did not observe any loci out of Hardy–Weinberg equilibrium. Levels of expected heterozygosity (HE) ranged from medium to high for all markers (mean HE = 0.79, and mean HO = 0.72). In the majority, there was no evidence of null alleles, but three loci (Spar05, Spar07, Spar013) showed significant frequencies of null alleles (above 15%–Table 2).

Table 2 Diagnostic characteristics of selected microsatellites.

Number of alleles, size range, polymorphic information (PI), observed heterozygosity (HO), expected heterozygosity (HE), Hardy–Weinberg equilibrium (HWE), and null alleles.

Locus	GenBank accession number	No. alleles	Size range (bp)	PI	HO	HE	HWE	Null alleles	
Spar01	KY645946	7	132–236	0.7098	0.941	0.761	0.08	×	
Spar02	KY645947	6	130–222	0.6455	0.765	0.692	0.08	×	
Spar05	KY645948	6	124–226	0.6069	0.412	0.699	0.05	✓	
Spar07	KY645949	10	121–226	0.8028	0.824	0.865	0.18	✓	
Spar08	KY645950	11	130–382	0.8052	0.800	0.860	0.13	×	
Spar09	KY645951	13	134–230	0.8864	0.875	0.933	0.11	×	
Spar010	KY645952	12	132–236	0.8698	0.882	0.919	0.08	×	
Spar011	KY645953	8	124–222	0.8125	0.588	0.863	0.12	×	
Spar012	KY645954	8	128–214	0.7068	0.750	0.772	0.08	×	
Spar013	KY645955	10	124–220	0.8577	0.500	0.867	0.05	✓	
Spar030	KY645957	6	133–169	0.7088	0.741	0.735	0.08	×	
Spar040	KY645958	6	124–190	0.6721	0.662	0.669	0.08	×	

Cross-species amplification showed differences for the twelve related species (Table 3). S. new species 3 presented the largest number of amplified microsatellites (8), followed by S. bakeri (7). S. mordax had the lowest number of amplified loci (4).

Table 3 Cross-species amplifications of the designed primers for S. parvidens.

We followed same PCR conditions in the twelve related species. (×) no positive amplification, (✓p) positive polymorphic amplification, (✓m) positive monomorphic amplification, (✓*) polymorphism not proven because PCR conditions were not standardized.

Locus	S. hondurensis (n = 3)	S. burtonlimi (n = 3)	S. oporaphilum (n = 1)	S. mordax (n = 2)	
Spar01	×	✓p	×	×	
Spar02	✓p	×	✓*	×	
Spar05	✓p	✓p	✓*	✓*	
Spar07	×	×	×	×	
Spar08	✓*	✓p	✓*	✓p	
Spar09	×	✓p	✓*	✓*	
Spar010	×	✓*	✓*	×	
Spar011	✓*	✓p	✓*	✓p	
Spar012	✓m	×	✓*	×	
Spar013	×	×	✓*	×	
S. tildae (n = 1)	S. erythromos (n = 1)	S. magna (n = 1)	S. bogotensis (n = 1)	S. newspecies_3 (n = 3)	S. luisi (n = 3)	S. lilium (n = 3)	S. bakeri (n = 2)	
×	×	×	×	✓p	×	✓*	✓*	
×	×	✓*	×	✓*	×	×	✓*	
✓*	×	✓*	×	✓*	✓*	✓*	×	
×	×	×	×	✓p	×	×	✓p	
✓*	✓*	×	✓*	✓p	✓*	✓*	✓p	
✓*	✓*	×	✓*	✓p	✓*	✓*	✓p	
✓*	✓*	✓*	✓*	✓p	✓*	✓p	✓p	
✓*	✓*	✓*	✓*	✓*	✓p	✓*	✓*	
✓*	✓*	✓*	✓*	×	×	×	×	
✓*	✓*	✓*	✓*	×	×	×	×	

Discussion

Next Generation Sequencing allowed the project to obtain a large number of microsatellite loci for Sturnira parvidens. This method has been probed for several bat species, and it is becoming a standard method for acquiring specific molecular markers (McCulloch & Stevens, 2011). Given the natural applicability of microsatellites to solve ecological questions, these molecular markers have emerged as a multipurpose indicator for ecological applications (Zane, Bargelloni & Patarnello, 2002; Selkoe & Toonen, 2006). Its applicability spreads to different academic fields such as population genetics, behavioral ecology, genomics, phylogenies, etc.

Our microsatellites conformed to the normal standard measures (Balloux & Lugon-Moulin, 2002). These indicators provide a straightforward approach for describing genetic variation due to the high level of existing alleles. Low allelic richness can affect accuracy in estimating population genetic parameters, leading to significant errors in assessing genetic diversity of target populations (Bashalkhanov, Pandey & Rajora, 2009). Here, we present a novel set of microsatellite loci with the potential to estimate genetic diversity in a non-model species. Standard measures for our microsatellites may have important implications in the evolutionary biology of the target species, because they can be used to develop conservation strategies for Neotropical bats. Highly informative microsatellites have been used to assess genetic diversity in a broad range of bat populations and to propose measures for conservation (i.e., Rossiter et al., 2000; Romero-Nava, León-Paniagua & Ortega, 2014; Korstian, Hale & Williams, 2015).

Amplified microsatellites for S. parvidens presented levels of polymorphism and heterozygosity similar to those found in other bat species (i.e., Artibeus jamaicensis—Ortega et al., 2002; Rhinolophus ferrumequinum—Dawson et al., 2004; Desmodus rotundus-Piaggio, Johnston & Perkins, 2008; Corynorhinus spp.—Lee, Howell & Van Den Bussche, 2011; Myotis spp.—Jan et al., 2012; Carollia castanea—Cleary, Waits & Hohenlohe, 2016).

Microsatellite markers are widely used to infer levels of genetic diversity in natural populations. Molecular markers are not always developed for the target species and the use of microsatellite loci from related species can be accurate. Ascertainment bias limited the microsatellite-based amplification due to the particular selection of polymorphic markers in the target species, plus the reduced sensitivity of the markers due to the phylogenetic constrictions of the particular evolutionary traits of each sister species (Crawford et al., 1998; Schlötterer, 2000; Li & Kimmel, 2013). The bias leads to a lower average allele length due to the phylogenetic restriction provided by the unique evolutionary history of each species (Li & Kimmel, 2013). We tested the potential use of our markers in related species, finding multilocus heterozygosities inside the Sturnira genus. This positive effect suggests the use of the developed markers to extrapolate genetic diversity in future studies for this highly speciose genus, in which the past demographic shared histories barely affect the cross-species amplification consolidation.

Conclusions

We used Illumina Paired-Sequences to efficiently develop microsatellite loci for Sturnira parvidens. We formed a genomic library to obtain 12 specific and polymorphic microsatellites for this bat. Microsatellites showed high allelic richness per locus, showing their effectiveness for further studies (i.e., population genetics, behavioral ecology, etc.). Cross-species amplification was effective for the 12 related species, but with no positive amplifications in several cases.

Supplemental Information

Supplemental Information 1 Supplementary material

Click here for additional data file.

Supplemental Information 2 Raw data in a fasta file

Click here for additional data file.

Supplemental Information 3 Raw data

Click here for additional data file.

We are grateful for the supporting fieldwork provided by students of the Facultad de Ciencias, UNAM (MZFC-M). We would like to thank the Field Museum of Natural History, Chicago (FMNH); Louisiana State University, Lousiana State University, Museum of Zoology, Baton Rouge (LSUMZ); Museo de Zoología de la Universidad de Costa Rica, San José, Costa Rica (MZUCR); Museum of Texas Tech University, Lubbock (TTU) for providing tissues from their collections. Particular special thanks go to Bruce D. Patterson and Natalia Cortés-Delgado from FMNH; David Villalobos and Bernal Rodríguez from MZUCR; Frederick H. Sheldon and Donna L. Dittmann from LSUMZ; and Caleb D. Phillips from TTU for providing samples.

Additional Information and Declarations

Competing Interests

Author Contributions

Field Study Permissions

DNA Deposition

Data Availability

The authors declare there are no competing interests.

Edgar G. Gutiérrez and Giovani Hernández Canchola conceived and designed the experiments, analyzed the data, wrote the paper, prepared figures and/or tables, reviewed drafts of the paper.

Livia S. León Paniagua and Jorge Ortega conceived and designed the experiments, contributed reagents/materials/analysis tools, wrote the paper, reviewed drafts of the paper.

Norberto Martínez Méndez conceived and designed the experiments, analyzed the data, wrote the paper, reviewed drafts of the paper.

The following information was supplied relating to field study approvals (i.e., approving body and any reference numbers):

Secretaría de Medio Ambiente y Recursos Naturales (SEMARNAT) provided permit for tissue collection FAUT-0307.

The following information was supplied regarding the deposition of DNA sequences:

Gen Bank: Spar01 KY645946, Spar02 KY645947, Spar05 KY645948, Spar07 KY645949, Spar08 KY645950, Spar09 KY645951, Spar010 KY645952, Spar011 KY645953, Spar012 KY645954, Spar013 KY645955, Spar030 KY645957, Spar040 KY645958.

The following information was supplied regarding data availability:

The raw data has been supplied as a Supplementary File.

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
