# Peer review of "Isolation and characterization of microsatellite markers for Sturnira parvidens and cross-species amplification in Sturnira species"

_PeerJ, doi:10.7717/peerj.3367_

## Round 0.1 · original submission · Minor Revisions

· Academic Editor

Minor Revisions

This paper will be acceptable for publication in PeerJ once all of the concerns of the myself and the reviewers are addressed. The authors should aim to improve the clarity of the manuscript by making improvements to the English, adding additional details of the methods, and writing a focused discussion that specifically addresses the results presented. While I believe that PeerJ has some flexibility in the paper structure, separating the Results from the Discussion may help the authors improve the focus and organization of these sections. I have also attached an annotated pdf with highlighted words/sentences for revision and associated comments. If the authors would like to include the sequences of each locus in addition to the genbank accession numbers, they should be in a standard format for re-use, such as fasta.

Reviewer 1 ·

Basic reporting

- Last month one new species of Sturnira was described. Should be added to the introduction.
"Molinari, J., Bustos, X.E., Burneo, S.F., Camacho, M.A., Moreno, S.A. & Fermín, G. 2017. A new polytypic species of yellow-shouldered bats, genus Sturnira (Mammalia: Chiroptera: Phyllostomidae), from the Andean and coastal mountain systems of Venezuela and Colombia. Zootaxa 4243(1): 75–96."

I believe articles "submitted" cannot be cited in PeerJ. Because of this "Hernández-Canchola & León-Paniagua, submitted" should be removed from the text.

Experimental design

no comment

Validity of the findings

no comment

Comments for the author

This is an informative manuscript. It reports microsatellite markers for a bat of the genus Sturnira. The methodology used to generate and analyze the data is appropriate and it provides an interesting first look at microsatellite diversity for this genus.

Reviewer 2 ·

Basic reporting

Overall statement: The manuscript is describing the development of microsatellites for a Neotropical bat species, Sturnira parvidens, using a well-established method to identify and test the microsatellite loci of the bat species for further molecular ecology studies. They have also provided limited further tests of the usefulness of these new markers for congeneric species. Overall, I find this paper to provide a useful technical contribution for population genetic research of Sturnira parvidens. However, there are substantial problems with this paper, largely with respect to the quality of the writing, largely in terms of professional writing style and English language and expression. The structure of the manuscript does not conform to PeerJ’s standards. Further, some results are missing (most notably, the authors state that cross-amplification was trialled in 12 species, results for only 9 are shown). I cannot recommend this paper for publication unless substantial revisions are made.

1. The English language should be improved to ensure that the paper is clear, understandable and professional. There are major issues including grammatical errors, tense changes within and between sentences, mistakes in conjugation and word choice, and ambiguity of meaning. I would recommend having a native speaker colleague or colleague more proficient in writing in English review the manuscript. You may also wish to consider language editing services, some of which offer discounts for researchers from lower-resource settings. Some examples of poorly formulated sentences and ambiguity include line 45-46, 53-55, 69-70,119-120,136-137.

2. The structure does not adhere to PeerJ’s described standards. Results and Discussion should be presented as separate sections. The Conclusion should be presented with complete sentences and paragraph(s), not as bullet points.

3. The title could be clearer about what was actually done (e.g. "Isolation and characterisation of....", and it is clearer and reads better if the methods are either described more actively (i.e. "...using Illumina paired-end sequencing" rather than putting this in parentheses). However as this method is fairly standard, I suggest simply excluding the reference to Illumina paired-end sequencing in te title. I suggest changing to “Isolation and characterisation of microsatellite markers for Sturnira parvidens and cross-species amplification in Sturnira species", or similar.

4. The authors describe cross-amplification of the developed markers in 12 bat species in the abstract and methods and conclusions, however the results in Table 4 show only 9 species.

Further, the authors describe levels of polymorphism, without stating the number of individuals from which the microsatellites were amplified for each species. Please amend this. If only one specimen was used for each species trial, it would be clearer to simply state whether than indivial was homozygotic or heterozygotic for each locus.

5. The background information provided in the introduction is interesting, but may be a little excessive for what is essentially a primer note. I would recommend cutting it down and limiting much of the first two paragraphs to highlighting knowledge gaps that the use of these markers may help to address.

6. Figure 1 – this is a nice distribution map, however, it is not relevant for this particular paper. Recommend cutting this from the manuscript.

7. Table 1 – this is more a figure than a table. The figure is redundant as it provides no additional information compared to the data lines below, suggest presenting as a table only. Also suggest considering including this as a supplementary table rather than in the paper proper.

8. Table 2. Please clarify the meaning of “x” as “failed to amplify” in a footnote so that the table stands alone.

9. Table 3. Please provide the precise p-values rather than >0.05. Please provide the null allele frequency estimate, as opposed to ticks and crosses. Please provide the number of individuals for which the microsatellites were amplified either in the title, or if it varies (ie if some failed to amplify for some loci), as a separate column.

Please change the header "Alleles range" to "Size range (bp)" or "Allele size range (bp)".

10. Table 4. Thank you for this useful table that will be of use for the study of multiple species within the genus. Please place the title above the table.

11. Line 117-118 – this sentence is contradictory – there was no evidence of null alleles, but three loci showed significant frequencies of null alleles?

12. Thank you for submitting the raw microsatellite sequences, but please provide them in a more usable format for dna sequences (e.g. fasta). In order to fulfil PeerJ's requirement for raw data, please provide examples of scored peaks for each of the microsatellite loci.

13. Line 58 “very little is known about microsatellite markers” - Markers don’t become known, they are developed.

14. Revise wording in lines 59-63 to be more precise and specific. E.g. that microsat markers provide a powerful tool to investigate questions on human impacts and habitat fragmentation by examining dispersal and population connectivity, and to better understand mating behaviour and social systems through parentage analysis (this is just a guide/example sentence).

15. Please replace all uses of the word “etc” as this is not appropriate for a scientific paper

16. Meaning of the last sentence of the introduction (lines 65-66) is unclear, please clarify.

17. The meaning of the sentence in lines 69-70 “Matters are proportionate by…” is unclear, please clarify.

18. Please clarify what is meant by “for use of wild” (line 71), and what kind of guidelines were followed.

19. Lines 75-76 – please clarify what kind of extraction method was used for the samples sent for shotgun sequencing (i.e. only Qiagen-kit extracted or both methods)

20. Line 80, PAL_FINDER version number only needs to be stated once

21. Line 81 please provide reference and version for MSATCOMMANDER

22. Line 90-91 - there’s no need to describe using an excel spreadsheet and csv file, please remove this.

23. Repetition of ideas presented in lines 49-51 and the following paragraph, I suggest cutting the former

24. Line 139 please clarify the intended meaning of “minor average allele length due to the phylogenetic restriction”

25. Supplement 1: Please provide museum names, and clarify your intent for the last column / unify the contents. “Polimorfism” is not a use of the tissue. Please clarify what you mean.

26. Supplement 2: This figure is incredibly difficult to read in tiny font size and light grey font colour. But my biggest issue with this figure is the pie charts, which are hard for the brain to interpret, particularly when presented in 3D. Please change these to stacked bar chart or similar.

27. Line 103 please clarify the phrase “were the less abundant in our lecture”

28. Line 104 – annealing temperature should be in the methods, this is not a result

29. Line 38 “they like to…” is informal and subjective. I suggest changing this to “… are commonly found…”

30. Please submit the manuscript in portrait, not landscape form, for the benefit of the reviewer/editor.

31. Line 47 add reference after the first sentence mentioning of another piece of research.

32. Line 49 – speciose not specious

33. Table 4. The symbols used are busy and make it difficult to read. I suggest considering replacing the ticks with subscript to letters (e.g. just use p for positive polymorphic, m for positive monomorphic, etc) (just a suggestion)

Experimental design

Experimental design appears to be of a high standard. Methods are generally of sufficient detail of replication. Some information is missing from the methods, such as the number of individuals used for cross-amplification.

34. Please state the brand and manufacturer of the genetic analyser used for electrophoresis of the PCR products to produce the microsatellite traces.

35. Line 82 although your methods are described generally thoroughly, there is no justification for the 14 loci chosen for PCR trials. Please provide a brief justification.

36. Line 83-84 – was magnesium chloride used in the assays?

Validity of the findings

Data are robust, findings are valid, with the exception of the missing results for three cross-amplified species.

Conclusions are ok in content, again with the exception of the missing results, but reporting fails to meet standards (reported results and discussion together, conclusion written in point form with incomplete sentences, see section 1).

Comments for the author

The manuscript is describing the development of microsatellites for a Neotropical bat species, Sturnira parvidens, using a well-established method to identify and test the microsatellite loci of the bat species for further molecular ecology studies. They have also provided limited further tests of the usefulness of these new markers for congeneric species. Overall, I find this paper to provide a useful technical contribution for population genetic research of Sturnira parvidens. However, there are substantial problems with this paper, largely with respect to the quality of the writing, largely in terms of professional writing style and English language and expression. The structure of the manuscript does not conform to PeerJ’s standards. Further, some results are missing (most notably, the authors state that cross-amplification was trialled in 12 species, results for only 9 are shown). I cannot recommend this paper for publication unless substantial revisions are made.

To the authors, I think the quality of the experimental work here is good, there are just some oversights in terms of reporting, particularly with those missing results, and some structure and language issues. This detracts from the quality of the scientific work performed. If these issues are addressed this will be a fine primer note and a useful resource.

---

## Round 0.2 · accepted · Accept

· Academic Editor

Accept

Thank you for your revision. There remain several statements in the manuscript that could use additional clarification. I have attached an annotated pdf for you to consider as you send your manuscript to production. Also, the raw data was not re-uploaded in fasta format as was stated in the response to reviewers, please fix this.